# Single-dose bNAb cocktail or abbreviated ART post-exposure regimens achieve tight SHIV control without adaptive immunity

Mariya B. Shapiro [1], Tracy Cheever[2], Delphine C. Malherbe [2,8], Shilpi Pandey[2], Jason Reed[3], Eun Sung Yang[4], Keyun Wang[4], Amarendra Pegu [4], Xuejun Chen [4], Don Siess[5], David Burke[5], Heidi Henderson[2], Rebecca Lewinsohn[2], Miranda Fischer[2], Jeffrey J. Stanton[6], Michael K. Axthelm [2], Christoph Kahl[5,9], Byung Park[7], Anne D. Lewis [6], Jonah B. Sacha[1,2,3], John R. Mascola[4], Ann J. Hessell[2] & Nancy L. Haigwood[1,2]*

Vertical transmission accounts for most human immunodeficiency virus (HIV) infection in children, and treatments for newborns are needed to abrogate infection or limit disease progression. We showed previously that short-term broadly neutralizing antibody (bNAb) therapy given 24 h after oral exposure cleared simian-human immunodeficiency virus (SHIV) in a macaque model of perinatal infection. Here, we report that all infants given either a single dose of bNAbs at 30 h, or a 21-day triple-drug ART regimen at 48 h, are aviremic with almost no virus in tissues. In contrast, bNAb treatment beginning at 48 h leads to tight control without adaptive immune responses in half of animals. We conclude that both bNAbs and ART mediate effective post-exposure prophylaxis in infant macaques within 30–48 h of oral SHIV exposure. Our findings suggest that optimizing the treatment regimen may extend the window of opportunity for preventing perinatal HIV infection when treatment is delayed.

[1] Department of Molecular Microbiology and Immunology, Oregon Health & Science University, Portland, OR, USA. [2] Division of Pathobiology & Immunology, Oregon National Primate Research Center, Oregon Health & Science University, Beaverton, OR, USA. [3] Vaccine & Gene Therapy Institute, Oregon Health & Science University, Beaverton, OR, USA. [4] Vaccine Research Center, NIAID/NIH, Bethesda, MD, USA. [5] Molecular Virology Core, Oregon National Primate Research Center, Oregon Health & Science University, Beaverton, OR, USA. [6] Division of Comparative Medicine, Oregon National Primate Research Center, Oregon Health & Science University, Beaverton, OR, USA. [7] Biostatistics & Bioinformatics Core, Oregon National Primate Research Center, Oregon Health & Science University, Beaverton, OR, USA. [8] Present address: University of Texas Medical Branch, Galveston, TX, USA. [9] Present address: Atara Biotherapeutics, Thousand Oaks, CA, USA. *email: haigwoon@ohsu.edu

A hallmark of human immunodeficiency virus type 1 (HIV-1) infection is the early establishment of a persistent viral reservoir[1]. Daily antiretroviral therapy (ART), the current standard of care, can reduce plasma viral load (PVL) to undetectable levels, but treatment interruption results in viral rebound. Nonetheless, post-exposure prophylaxis (PEP) with ART within 72 h of exposure can reduce the likelihood of infection, although efficacy wanes the longer treatment is delayed[2]. Time of exposure can be determined with relative accuracy in the setting of mother-to-child transmission (MTCT), where over 180,000 children acquire HIV-1 each year (http://www.unaids.org/en/resources/documents/2018/unaids-data-2018) and, without treatment, half die before age two[3]. Because transmission is most likely to occur late in pregnancy or peripartum[4], risk is minimized by treating mothers during pregnancy and newborns at birth[5]. Nonetheless, even in infants born to untreated mothers, infection risk was reduced by treatment with the ART drug zidovudine beginning within 48 h of birth, although treatment beginning at 3 days of age or later was ineffective[6]. Once virus is detected in the infant, lifelong ART is typically necessary to maintain suppression; however, there have been isolated cases of durable long-term remission in children following ART interruption[7–9]. The best-known case, the Mississippi baby, was already viremic when treatment began—evidence for infection in utero rather than peripartum[7]—and ultimately experienced viral rebound[8]. Thus, achieving remission in children perinatally infected with HIV may depend upon the time between earliest virus exposure and initiation of therapy; those infected in utero will likely be the most difficult to treat.

Nonhuman primate models of newborn and infant infection with simian immunodeficiency virus (SIV) and chimeric simian-human immunodeficiency virus (SHIV) have allowed in-depth investigations of viral dissemination and effectiveness of pre-exposure and post-exposure ART or broadly neutralizing antibody (bNAb) treatments[10]. SIV and SHIV disseminate rapidly in infant rhesus macaques, reaching distal tissues within one day of exposure[11,12]. Studies of early intervention using ART in both infant and adult macaques after SIV exposure have suggested that both timing and duration of treatment are critical for effective PEP[13–16]. Recently, Okoye et al. showed evidence for durable tight control in adult macaques treated with ART for 600 days beginning 4–5 days after SIV infection and then released from ART, while delaying initiation of treatment to 6 days or longer led to viral rebound[17].

ART has been indispensable in reducing vertical transmission of HIV-1, yet many children have no access to treatment, and adherence is hindered by drug regimen complexity, frequent dosing intervals, and poor palatability[18]. Suboptimal treatment raises the risk of drug resistance[19], underscoring the need for more accessible interventions. Passive treatment with bNAbs offers several theoretical advantages over ART, including longer half-lives, simpler treatment regimens, and the potential for infected cell killing by Fc-mediated innate immune system engagement[20]. Potent bNAbs block SHIV infection in nonhuman primate models[21,22] and suppress viremia transiently in humans treated during chronic HIV-1 infection[23,24]. The use of bNAbs in the setting of vertical transmission is currently under evaluation in clinical trials[25,26]. Similar to human infants exposed to HIV-1 in maternal blood and cervico-vaginal secretions at birth, one-month-old infant macaques exposed orally to high doses of SHIV$_{SF162P3}$ develop high viremia and rapidly progress to disease[12,27,28]. This SHIV stock is a 'swarm' of variants that is classified as Tier 2 in neutralization sensitivity[21]. In this model of perinatal infection, we reported full clearance of viral foci and prevention of reservoir establishment with short-term bNAb cocktail treatment when treatment was initiated 1 day following SHIV exposure[12]. In contrast, a delay in treatment to 10 days[29] or even 3 days[30] after mucosal SHIV exposure in adult macaques resulted in viral resurgence after bNAb decay. However, the window of opportunity for PEP using bNAbs in young animals has not been strictly defined. Because treatment of an HIV-exposed newborn may be delayed by hours or days in a clinical setting, it is imperative to determine the time window for PEP using bNAbs and to examine which variables have the greatest impact on efficacy in the context of delayed PEP.

In the present study, we test the effect of extending the interval before bNAb treatment initiation time to 30 h and 2 days post-exposure. Treatment with a single 40 mg/kg dose of bNAb cocktail at 30 h is highly effective, with no evidence of virus in tissues at necropsy. In contrast, treatment with a 4-dose regimen of 10 mg/kg bNAb cocktail beginning at 48 h results in durable control of plasma viremia in only half of treated animals. In comparison, we evaluate short-term ART beginning 2 days post-exposure in this model and find that a 21-day course beginning at 48 h is also highly effective, with no viral rebound observed after ART cessation and little to no detectable virus in tissues. Regardless of treatment regimen, tight controllers do not mount long-lived adaptive immune responses and do not become viremic after CD8$^+$ T cell depletion, suggesting viral control by other mechanisms. Taken together, our findings begin to define the variables that impact the window of opportunity for PEP in this setting and show that viral clearance with either bNAbs or ART is an achievable outcome.

## Results

**Study design.** Previously, we tested bNAbs PGT121[31] and VRC07-523[32] together as PEP in infant macaques. Cocktails of either 10 mg/kg (each bNAb at 5 mg/kg) or 40 mg/kg (each bNAb at 20 mg/kg), given subcutaneously on days 1, 4, 7, and 10, initiated 24 h after oral SHIV$_{SF162P3}$ exposure resulted in viral clearance and no evidence of persistent infection[12]. To test the impact of delaying bNAb treatment, we administered the same cocktail of bNAbs at 10 mg/kg (5 mg/kg each bNAb) subcutaneously to two groups of six infant macaques beginning 48 h after high dose oral SHIV$_{SF162P3}$ challenge and again on days 4, 7, and 10 (Fig. 1, Groups 2A and 2B). Animals in both groups received a SHIV challenge dose that reproducibly infected all untreated controls (Supplementary Table 1); however, a different SHIV challenge stock was used for Group 2A, an earlier pilot study. We also tested the outcome of a single bNAb cocktail dose at 30 h, a more pragmatic clinical regimen. Because PGT121 persists longer in vivo than VRC07-523[12], and because mismatched decay kinetics of bNAbs in a cocktail have been shown to promote viral escape when only one bNAb remains[24], we replaced VRC07-523 with VRC07-523LS[32], which has mutations in the Fc at positions M428L and N434S that confer a longer half-life in vivo[33]. Six animals were exposed to SHIV orally followed by a single subcutaneous dose of bNAb cocktail containing PGT121 and VRC07-523LS at 40 mg/kg (20 mg/kg each bNAb) (Fig. 1, Group 3). For comparison with the bNAb treatments, we exposed 6 animals to the same SHIV dose orally and treated with an ART cocktail of emtricitabine (FTC, 40 mg/kg/day), tenofovir disoproxil fumarate (TDF, 5.1 mg/kg/day), and dolutegravir (DTG, 2.5 mg/kg/day)[34] using an abbreviated 21-day regimen. The drugs were administered daily by subcutaneous injection beginning at 48 h and continuing through day 23 (Fig. 1, Group 4). Five age-matched animals exposed to the same SHIV dose were used as contemporaneous controls alongside Groups 2B, 3, and 4 (Fig. 1, Group 1).

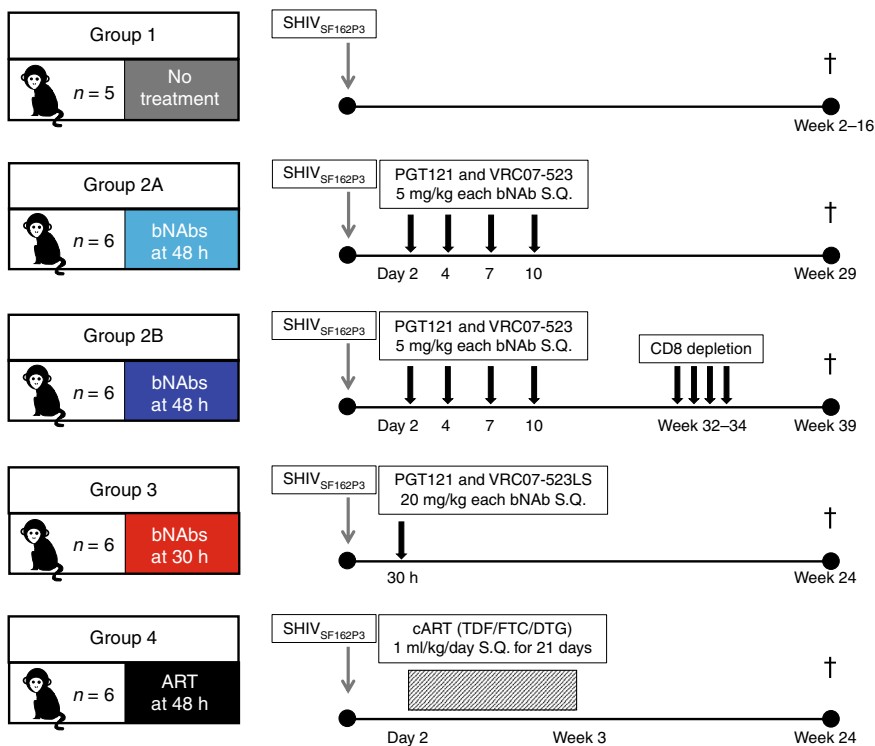

**Fig. 1 Study design.** One-month-old infant rhesus macaques negative for *Mamu-B\*08* and *Mamu-B\*17* alleles were randomly assigned to groups. All animals were exposed to a single high dose SHIV$_{SF162P3}$ challenge by atraumatic oral inoculation. S.Q., subcutaneous delivery. TDF, tenofovir disoproxil fumarate (5.1 mg/kg). FTC, emtricitabine (40 mg/kg). DTG, dolutegravir (2.5 mg/kg). Gray arrows indicate SHIV infection. Black arrows indicate bNAb treatments. Rectangle with diagonal stripes indicates duration of daily ART treatment. Dagger indicates time of necropsy. Group colors are consistent throughout the manuscript. The number of animals (N) is indicated next to the monkey cartoon in each group.

**Durable control of plasma viremia after bNAbs or ART.** To compare the virologic outcomes of delaying bNAb treatment to 30 h (Group 3) or 48 h (Groups 2A and 2B), or short-term ART initiated at 48 h (Group 4), we followed the animals for 24–39 weeks after SHIV exposure and quantified viral RNA in plasma and viral DNA (vDNA) in PBMC. Initiation of bNAb treatment at 48 h post-exposure (Group 2B) resulted in a delay in acquisition of sustained plasma and PBMC viremia compared with untreated controls (Fig. 2a–d). In the 3/6 animals (50%) with breakthrough viremia, persistent viremia was first detected 5–6 weeks post-exposure, compared with 4–7 days in controls. The remaining 3/6 (50%) animals suppressed viremia to levels below detection in plasma and PBMC, except for isolated transient blips of viremia in plasma. Similar outcomes were observed in the pilot study (Group 2A) (Supplementary Fig. 1a, b), but the corresponding control group of 2 animals precluded statistical analyses. The outcomes of full clearance when short-term treatment began at 24 h[12] and viral suppression in 50% of infants when treatment began at 48 h prompted us to test an intervening treatment time. A single-dose bNAb treatment at 30 h (Group 3) resulted in a lack of detectable viremia in both plasma and PBMC of 6/6 animals, except for isolated blips at early time points (Fig. 2e, f). These findings imply that a single high dose of bNAbs can prevent sustained viremia when administered as late as 30 h after SHIV exposure.

To better understand the incomplete prevention of viremia by bNAb treatment at 48 h, we asked whether viral breakthrough was due to escape mutations in the virus. Of the three animals in Group 2A that had breakthrough viremia, two (34215 and 34232) seroconverted 3–5 weeks after breakthrough viremia was first detected, while the third (34245) did not seroconvert (Supplementary Fig. 1c). To determine whether emerging viruses in these

animals were bNAb escape variants, we used single genome amplification (SGA) to clone full length *env* variants from plasma sampled shortly after viral breakthrough. Pseudoviruses made with these individual *env* clones were tested for sensitivity to neutralization by PGT121 and VRC07-523 (Supplementary Fig. 1d). Several clones obtained by SGA from the SHIV$_{SF162P3}$ stock used in this experiment were also included in the analyses. There was no evidence of any bNAb-resistant clones in either the stock or the plasma virus, suggesting that the bNAb cocktail could suppress virus expression, but that upon antibody decay virus grew out from reservoirs that had been seeded prior to bNAb therapy.

To determine whether control of viremia in bNAb-treated animals was mediated by CD8$^+$ T cells, we depleted CD8$^+$ cells in two moderately-viremic and two transiently-viremic animals in Group 2B. Anti-CD8α antibody was administered beginning at week 32 (36557, 36566) or week 34 (36494, 36505) after virus challenge (Fig. 2c, d). CD8$^+$ T cells were undetectable in blood within one week of the first depleting antibody dose and remained below 200 cells/µL in all 4 animals until necropsy (Supplementary Fig. 2). In the moderately-viremic animals, plasma viremia spiked within one week, increasing by 0.75 log$_{10}$ and 2.77 log$_{10}$ in 36505 and 36557 respectively. Viremia in 36505 decreased as CD8$^+$ T cells began to rebound, while viremia in 36557 remained high, coincident with a profound lack of CD8$^+$ T cell rebound in this animal. In contrast, viremia did not increase in either of the animals with transient viremia (Fig. 2c). These data suggest that CD8$^+$ T cells play a role in controlling moderate viremia but are not required for maintaining tight control.

We next investigated treatment with a 3-week course of ART beginning at 48 h after SHIV exposure (Group 4). In light of the incomplete viral blockade observed in animals treated with

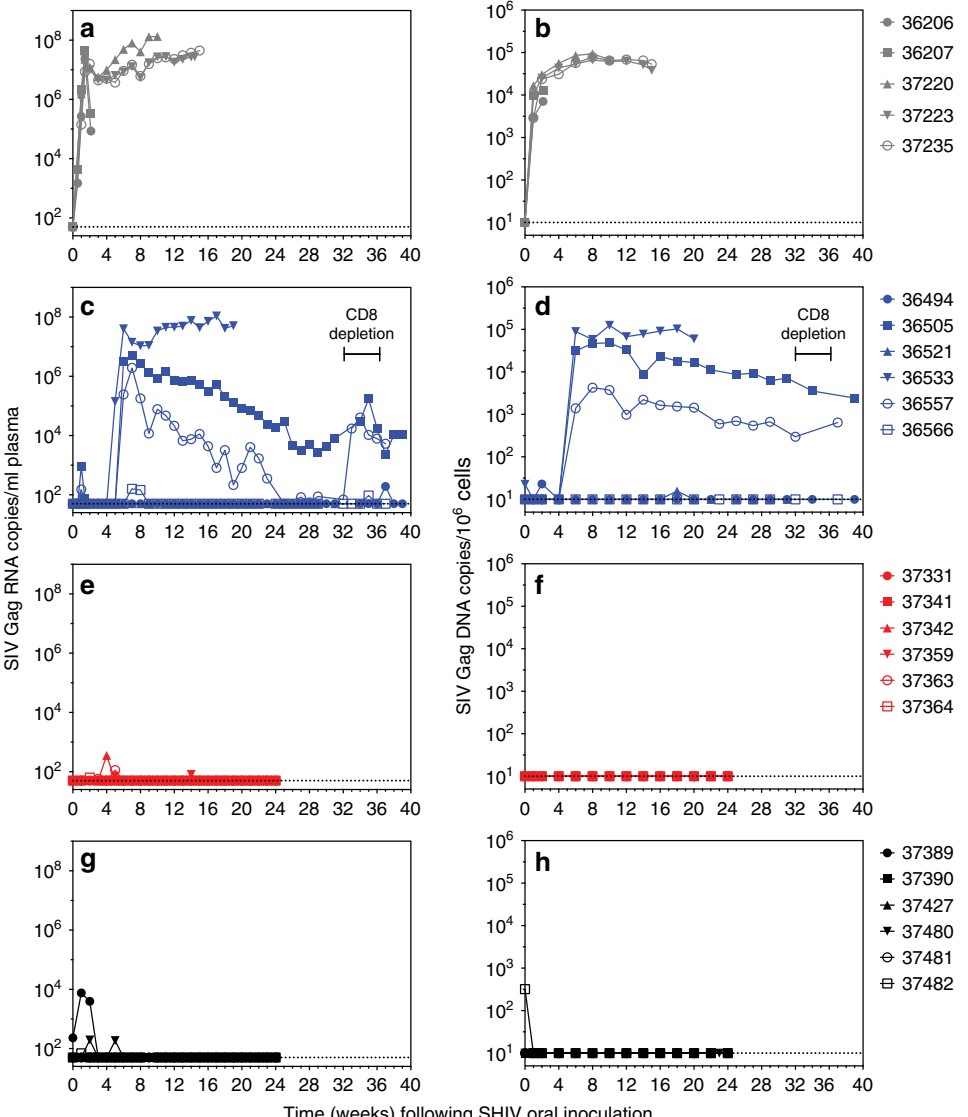

**Fig. 2 Viremia is attenuated by bNAbs or ART post-exposure.** Plasma viral loads (**a**, **c**, **e**, **g**) and PBMC-associated viral loads (**b**, **d**, **f**, **h**) were measured longitudinally. **a**, **b** Group 1 (untreated controls). **c**, **d** Group 2B (bNAbs at 48 h). CD8α depletion was performed on two tight controllers (36494, 36566) and 2 viremic animals (36505, 36557) during the time frame indicated. **e**, **f** Group 3 (bNAbs at 30 h). **g**, **h** Group 4 (ART at 48 h). Group colors and individual animal symbols are consistent throughout the manuscript. Source data are provided as a Source Data file.

bNAbs at 48 h, we hypothesized that this ART regimen would be less effective at clearing the nascent reservoir and preventing viral breakthrough. Unexpectedly, 0/6 animals experienced breakthrough after ART was discontinued (Fig. 2g, h). Two animals, 37389 and 37480, had detectable plasma virus in the first 2–5 weeks, but subsequently controlled viremia (Fig. 2g). For one animal, 37482, a low level of vDNA was detected in PBMC on day 0 prior to SHIV challenge, likely due to cross-contamination, but sample volume limitations prevented retesting (Fig. 2h).

To determine the efficacy of each of the treatments relative to the untreated controls, we used Kaplan-Meier analysis to quantify the statistical differences between the number of animals that became persistently viremic in each treatment group compared with the control group. Compared with untreated controls (Group 1), persistent viremia occurred in a significantly smaller proportion of animals that received bNAbs at 30 h (Group 3) or ART at 48 h (Group 4) (Log-Rank test, adjusted $p = 0.0002$ for both comparisons, Supplementary Fig. 3). While fewer animals became persistently viremic after receiving bNAbs at 48 h (Group

2B) compared with the control group, the difference was not statistically significant (Log-Rank test, adjusted $p = 0.0969$). In summary, our findings indicate that bNAb or ART treatment initiated shortly after SHIV exposure can delay or prevent onset of sustained viremia.

**bNAb pharmacokinetics and anti-drug antibody (ADA) responses.** Given the relatively short elapsed time of 18 h between the times of bNAb initiation in Groups 2A and 2B (48 h) and Group 3 (30 h), the degree to which their virologic outcomes differed was remarkably large and warranted further investigation. We measured bNAb concentrations in plasma to determine whether the pharmacokinetics (PK) of the different human bNAb regimens contributed to this disparity. In Group 2B animals, which received four doses of each bNAb at 5 mg/kg, PGT121 concentrations in plasma reached an average peak of 86 μg/ml, peaking by day 14 and decaying to undetectable levels in most animals within 5–6 weeks of SHIV exposure, with an average

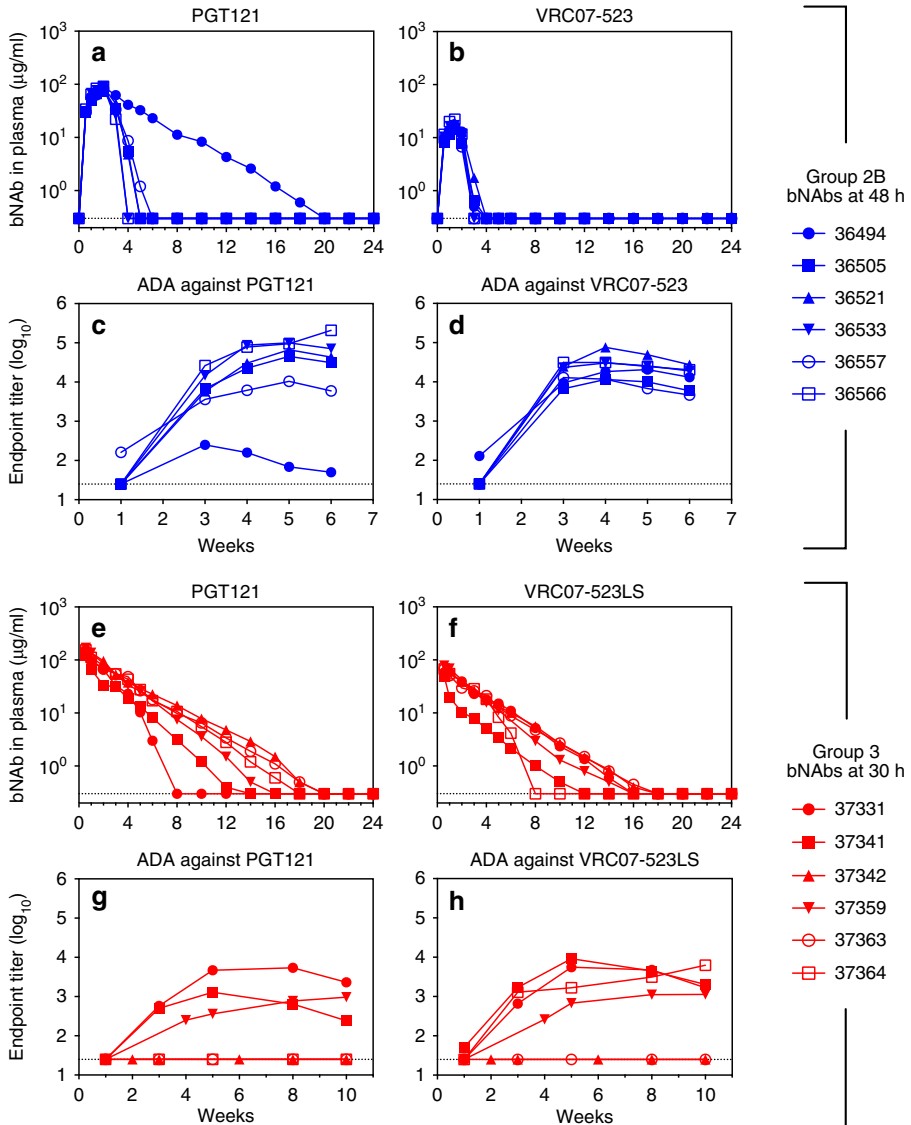

**Fig. 3 Pharmacokinetics of bNAb treatments. a**, **b** Plasma concentrations of PGT121 and VRC07-523 in Group 2B animals (5 mg/kg each bNAb at 48 h and on days 4, 7, and 10). Except for 36494, no animal had detectable PGT121 or VRC07-523 in plasma after week 6. **c**, **d** Endogenous anti-drug antibody (ADA) responses against passively administered bNAbs for Group 2B animals. **e**, **f** Plasma concentrations of PGT121 and VRC07-523LS in Group 3 animals (20 mg/kg each bNAb at 30 h). **g**, **h** Endogenous ADA responses against passively administered bNAbs for Group 3 animals. Concentrations of each bNAb were measured by ELISA using binding to ST0A9 (PGT121) or RSC3 (VRC07-523 and VRC07-523LS). Dotted lines indicate limits of detection. Group colors and individual animal symbols are consistent throughout the manuscript. Source data are provided as a Source Data file.

half-life of 2.4 days; the exception was animal 36494, in which PGT121 had a much longer half-life of 15.9 days and remained detectable up to week 18 (Fig. 3a). For VRC07-523, concentrations peaked on day 10, reaching an average of 17 µg/ml, and decayed within 3–4 weeks in all animals, with an average half-life of 2.9 days (Fig. 3b). Not surprisingly, clearance of these human bNAbs from the plasma was observed in conjunction with the development of robust ADA responses (Fig. 3c, d). Notably, 36494 had weaker ADA against PGT121, consistent with the lengthy persistence of this bNAb in this animal's plasma. Unexpectedly, both bNAbs showed greater persistence in Group 3 animals, which received a single 20 mg/kg dose of each bNAb. Plasma concentrations of both PGT121 and VRC07-523LS were highest on day 4 after SHIV exposure—the first day of measurement—and declined thereafter with half-lives of 12.4 and 12.9 days respectively. PGT121 reached an average concentration of 151 µg/ml and was detectable for 6–18 weeks (median 15 weeks) (Fig. 3e). Despite delivery at the same dose as PGT121,

VRC07-523LS peaked at only 65 µg/ml, yet was detectable for a similar length of time as PGT121 (range 6–16 weeks, median 14 weeks) (Fig. 3f). Because the bNAbs were given 30 h after SHIV exposure, it is possible that plasma bNAb concentrations peaked before day 4, prior to sampling. Of the six animals in this group, only three mounted ADA responses against PGT121 and four against VRC07-523LS, coincident with more rapid clearance of each bNAb in these animals (Fig. 3g, h).

To test whether the differences in peak bNAb concentration and bNAb half-life in vivo were determined by the treatment regimen, the specific bNAb, or both, we analyzed the data using a two-way ANOVA. Both bNAb ($p < 0.0001$) and treatment regimen ($p < 0.0001$) had a significant impact on the peak bNAb concentration in plasma; moreover, the interaction of bNAb and treatment regimen was significant ($p = 0.0395$), indicating that the treatment regimen affected the peak bNAb concentration differently for each bNAb (Supplementary Fig. 4a). For bNAb half-life, neither the bNAb nor the interaction of bNAb and

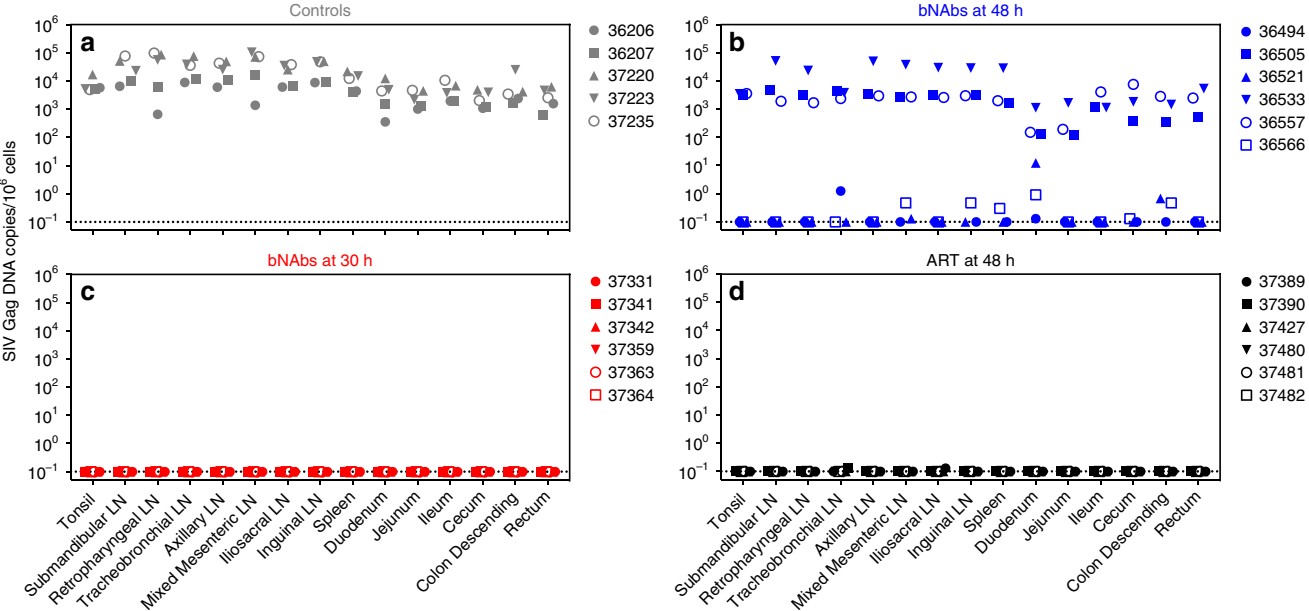

**Fig. 4 bNAb and ART treatments reduce virus in tissues.** Tissues were harvested at time of death and total viral DNA was quantified using an ultrasensitive SIV DNA qPCR assay. Total viral DNA quantities are shown for the tissue types listed along the bottom of the figure. **a** Group 1 (untreated controls). **b** Group 2B (bNAbs at 48 h). **c** Group 3 (bNAbs at 30 h). **d** Group 4 (ART at 48 h). All data shown is from tissues harvested at time of death. Group colors and individual animal symbols are consistent throughout the manuscript. See Supplementary Tables 2–6 for numerical data from these and other tissues.

treatment regimen were significant determinants. However, the single-dose treatment at 30 h (Group 3) resulted in a significantly longer half-life than the 4-dose treatment at 48 h (Group 2B) regardless of the bNAb ($p < 0.0001$) (Supplementary Fig. 4b).

**Quantitation of viral reservoir seeding in tissues.** We next asked how viral seeding in tissues was affected by post-exposure treatment with bNAbs or ART. Regardless of treatment, all persistently viremic animals—including 5/5 controls (Group 1) and 6/12 animals treated with bNAbs at 48 h (Groups 2A and 2B)—had moderate to high levels of vDNA in all tissues tested at necropsy. Lymphoid tissues, which included tonsil, lymph nodes, and spleen ($n = 9$ tissues per animal) generally harbored more vDNA than gastrointestinal (GI) tract tissues, which included duodenum, jejunum, ileum, cecum, descending colon, and rectum ($n = 6$ tissues per animal), by a factor of 6.9 (95% CI: 3.7–10.2, one-sample $t$ test, $p = 0.0015$, $n = 11$ animals) (Fig. 4a, b, Supplementary Tables 2–4). This difference probably reflects greater abundance of target cells in lymphoid tissues versus the GI tract. It is likely that much of the vDNA in the GI tract localized within gut-associated lymphoid tissues (GALT) such as Peyer's patches, but we did not dissect these tissues to isolate GALT specifically. In tight controllers—including 6/12 animals treated with bNAbs at 48 h (Groups 2A and 2B), and all animals treated with either bNAbs at 30 h (Group 3) or ART at 48 h (Group 4)—we detected no vDNA, or low levels in only a few tissues (Fig. 4b-d, Supplementary Tables 3–6). Inguinal lymph node biopsies at various time points throughout the study demonstrated that vDNA levels in this tissue, like those in PBMC (Fig. 2), remained stable over time in viremic animals and were undetectable in tight controllers (Supplementary Fig. 5). For animals in Groups 1, 2B, 3, and 4, which all received the same virus stock and challenge dose, the number of vDNA copies in lymphoid and GI tract tissues at necropsy was strongly correlated with the area under the PVL curve (AUC), as well as with peak PVL and with final PVL at necropsy (Supplementary Table 7). Because Group 2A animals

were challenged with a different stock, they were excluded from this analysis.

Assays that measure vDNA within tissues give a theoretical maximum size for the viral reservoir, but the majority of vDNA is likely to be defective genomes[35,36]. To better estimate the size of the replication-competent viral reservoir, we adapted the TZA assay, a reporter-cell-based quantitative viral outgrowth assay that is well-suited for quantifying inducible replication-competent virus in small samples, such as those obtainable from infant and juvenile macaques[37]. Of the three control group animals from which spleen samples were available, viral outgrowth from stimulated CD4+ cells was observed in two; the third was inconclusive due to microbial contamination (Supplementary Fig. 6). In contrast, inducible virus was not detected in the tissues of the 12 animals treated either with bNAbs at 30 h (Group 3) or with ART at 48 h (Group 4), except for one animal (37482) in Group 4 that had very low levels in the mesenteric lymph nodes. However, omission of TDF from the stimulation culture to permit de novo replication resulted in minimal but detectable outgrowth in several samples in Groups 3 and 4. In addition, a subset of animals treated with bNAbs at 48 h (Groups 2A and 2B) had inducible virus in spleen and/or mesenteric lymph nodes. These results suggest that early bNAb or ART therapy not only limits the tissue vDNA, but may also reduce the size of the inducible replication-competent reservoir in lymphoid tissues, although sample sizes in each group were insufficient for statistical comparison.

**Detection of adaptive immune responses to SHIV.** Previous reports have suggested that bNAb treatment beginning 3 days after SHIV exposure can promote the development of endogenous antiviral immune responses[30]. To assess the development of adaptive immunity in treated and control animals, we measured Env-binding IgG antibody titers in plasma, as well as T cell responses to Env, Gag, and/or Vif. Among the untreated animals, only 2/5 animals (40%) developed HIV-1 Env-specific antibodies, and binding was weak and transient (Fig. 5a), likely due to B cell

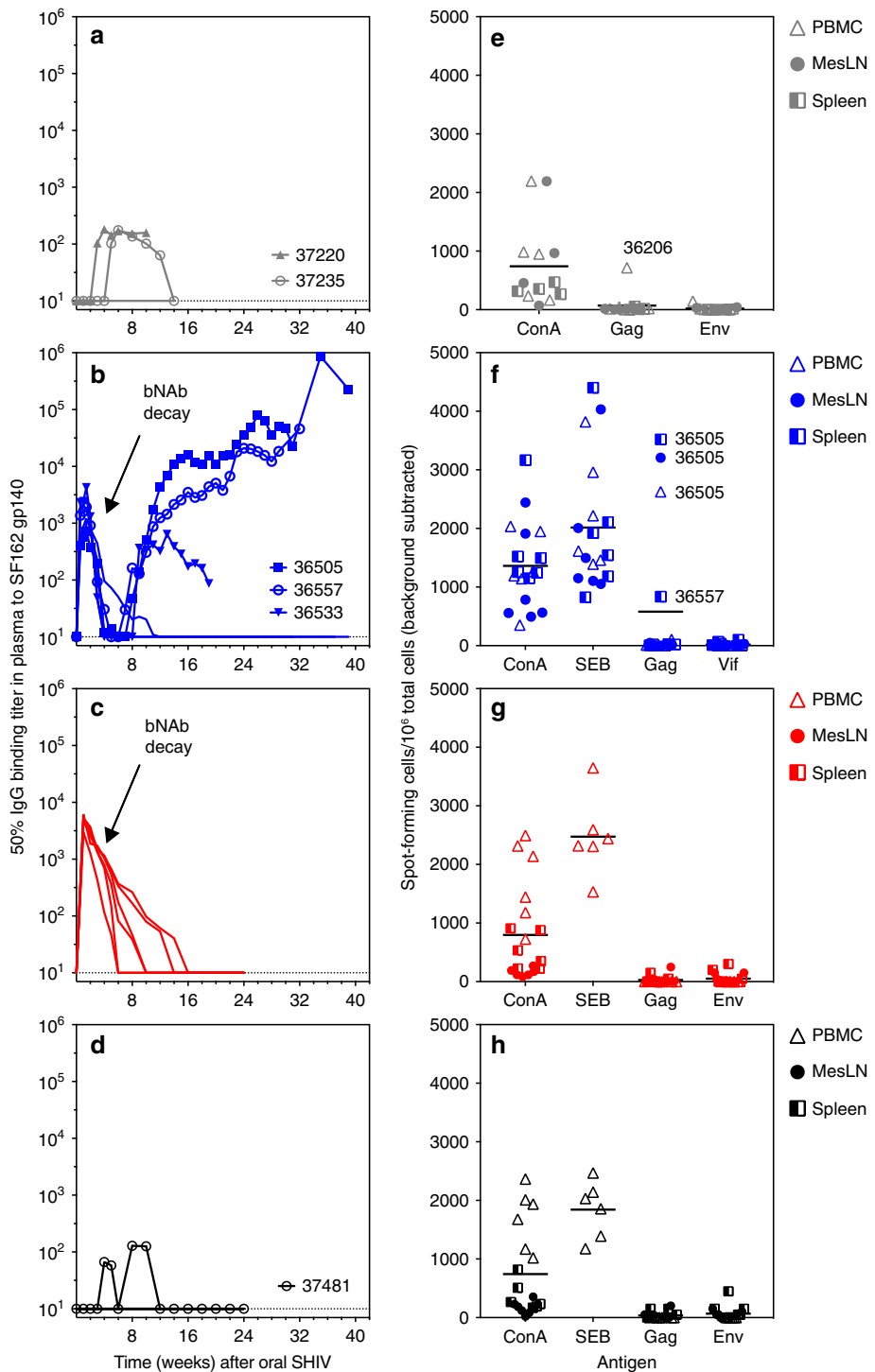

**Fig. 5 bNAbs and ART modulate adaptive immunity. a–d** Plasma IgG binding titers to SF162 Env protein (gp140) were measured by ELISA. Group colors and individual animal symbols are consistent throughout the manuscript. The animal number is indicated for animals that had positive antibody responses to SF162 gp140. **b**, **c** In Groups 2B and 3, the period of bNAb decay (rather than endogenous antibody response) is indicated by arrows. **e–h** T cell responses to HIV-1 Env antigen, SIVmac239 Gag ORF 15-mer peptide pool, or SIVmac239 Vif ORF 15-mer peptide pool were measured by IFNγ ELISPOT. SF162 gp140 protein was used as the Env antigen for all samples except for peripheral blood mononuclear cells (PBMC) for Groups 3 and 4 (**g**, **h**), for which Clade B consensus Env ORF 15-mer peptide pools were used. T cell responses in PBMC, mesenteric lymph nodes (MesLN), and spleen were quantified at time of death. Concanavalin A (ConA) and/or Staphylococcal Enterotoxin B (SEB) were used as positive stimulation controls. For each group, horizontal lines indicate the grand mean of ELISPOT responses to each antigen used for stimulation. Different tissue types (PBMC, MesLN, spleen) are indicated with individual symbols as shown. For specific tissue samples in which positive T cell responses to viral antigens were measured, the animal number is indicated next to the symbol. Group colors are consistent throughout the manuscript. Source data are provided as a Source Data file.

dysfunction resulting from uncontrolled viremia and pathogenesis[38,39]. In contrast, in the group treated with bNAbs at 48 h (Group 2B), moderately-viremic animals 36505 and 36557 developed persistent, increasing antibody responses, while highly viremic animal 36533 had transient seroconversion; the remaining three animals were tight controllers and did not seroconvert (Fig. 5b). In order to examine the contribution of ongoing B cell responses to viral control during chronic infection, the two animals with strongest seroconversion, 36505 and 36557, were given a single dose of anti-CD20 depleting antibody 25 and 23 weeks after SHIV exposure respectively; no changes in viral load (Fig. 2b) nor in plasma IgG binding titer (Fig. 5b) were observed. Because antibody-producing plasma cells and plasmablasts do not express CD20[40], and thus were not effectively depleted, this experiment was inconclusive. In animals treated either with bNAbs at 30 h (Group 3) or ART at 48 h (Group 4), transient or no seroconversion was observed, likely because none of these animals had a sufficient viral load to drive germinal center reactions (Fig. 5c, d). T cell responses were likewise detectable only in the moderately-viremic animals in Group 2B (Fig. 5f), and were weak or undetectable in both untreated animals with high viremia (Fig. 5e) and in bNAb- or ART-treated tight controllers (Fig. 5f-h). These data suggest that optimal adaptive immune responses depend on a moderate viral load, and that neither very high nor transient/undetectable viremia fosters B or T cell responses in this model.

**SHIV pathogenesis and clinical outcomes**. Finally, we evaluated pathology and clinical disease outcomes in treated and control infants (Supplementary Tables 8–12). Of the 5 untreated control animals (Group 1, Supplementary Table 8), two were sacrificed 15 days after SHIV exposure to confirm viral dissemination in tissues at this early time point (36206, 36207), and these did not have significant pathology. Of the remaining three controls, all had few to numerous opportunistic infections and lesions characteristic of SHIV disease. Among the viremic animals that received bNAbs at 48 h (Supplementary Tables 9–10), only the highly viremic animal 34245 (Group 2A, Supplementary Table 9) progressed to disease. Regardless of group or treatment, pathological outcomes of the aviremic animals were consistent with those from age-matched uninfected animals in the colony. No lesions were observed in animals with moderate and low viremia, indicating the absence of disease due to SHIV infection (Supplementary Tables 9–12). Regardless of treatment, peripheral CD4+ T cells did not decline precipitously in any animal over the course of the 6-month study timeline (Supplementary Fig. 7), consistent with observations that CD4+ counts and percentages remain elevated in human infants during ART treatment failure[41]. However, CD4+ counts were generally higher in animals with moderate and low viremia than in those with high and poorly controlled PVLs.

## Discussion

The ability to suppress viremia using ART has made HIV-1 a manageable chronic infection for patients with access to treatment. However, even intensified ART regimens cannot clear the viral reservoir, necessitating continuous therapy once persistent infection is established. In the case of the Mississippi baby, positive at birth and treated beginning at 30 h after birth, hopes for durable remission were dashed by the return of viremia[8]. Because this infant was likely infected in utero, the time between infection and treatment could not be determined. Recent and ongoing trials in newborns and children in the IMPAACT network as well as in young at-risk adults in the FRESH cohort[42] have focused on targeting primary viremia with ART as early as

1 day after the first positive viral test. The first such study in adults showed improvements in reducing both peak viremia and time to undetectable viremia, but it is not known if this has resulted in durable control[43]. In the present study, we have used a macaque model of perinatal HIV-1 infection to define the window of opportunity for intervention at known intervals after a single mucosal exposure with a high dose of SHIV that models exposure to a swarm of HIV-1 during birth and results in persistently high viremia and disease. Our data show that short-term therapy using either a single dose of bNAbs given within 30 h or a 21-day course of ART started 48 h after virus exposure can reliably prevent persistent viremia and limit vDNA in tissues. However, PEP with bNAbs initiated 48 h after SHIV exposure resulted in varying outcomes of reservoir establishment, shown by mitigated disease outcomes and tight control or clearance in half of the treated infants, similar to the frequency of control and clearance observed with a powerful T cell vaccine[44]. We used vDNA as a measure of the total virus because this is the most sensitive method for viral detection, and is standard for diagnosing HIV-1 infection in infants[45]. Testing of longitudinal lymph node biopsies showed that vDNA copy number was stable in individual animals over time, mirroring the data in PBMCs and consistent with viral measures of plasma viral RNA and outgrowth of inducible virus from tissues.

A key advantage of bNAbs over ART is the possibility of Fc-mediated cell killing. Two studies have suggested that direct killing of infected cells may be a mechanism of bNAb-mediated SHIV clearance[12,46]. In contrast, ART is unable to kill infected cells directly; nonetheless, its effect in vivo is rapid, limiting viral replication and spread[47]. We hypothesized that the use of bNAbs in combination with a short-term ART regimen could therefore be a more effective PEP approach than either treatment alone, and might extend the window of opportunity for intervention after HIV-1 exposure. Before testing bNAbs and ART in combination, we tested an abbreviated regimen of ART alone. Our present findings show a highly abbreviated ART regimen is effective when commenced 48 h after oral SHIV inoculation, suggesting that Fc-mediated killing of infected cells may not be required for bNAb PEP approaches. Instead, the data support a view that both ART and bNAbs act primarily by limiting viral replication and spread. The greater relative importance of neutralization over Fc function may be especially true in the case of highly potent bNAbs such as PGT121, which was able to protect monkeys from SHIV challenge despite abrogation of Fc receptor binding[48].

The finding that ART appeared to be more effective than bNAbs at the 48-h time point was unexpected. While the specific kinetics of reservoir seeding are likely virus-specific, our findings are generally consistent with studies showing that while ART is able to limit initial seeding, its effectiveness wanes rapidly as the reservoir is seeded in the first few days of SIVmac251 infection in adult macaques[16]. Moreover, we consider it likely that during the critical early window of rapid viral dissemination and reservoir seeding, the outcome of PEP depends as greatly on the speed and extent of drug biodistribution to the sites of viral replication as it does on the potency of viral blockade. Being small molecules, ART drugs rapidly diffuse to distal sites; for instance, DTG was detectable at therapeutically relevant concentrations in colorectal tissue within 1 h of a single oral dose in a Phase I trial[49]. In contrast, monoclonal antibodies are much larger, and their rate of convection across interstitial space is influenced by antibody size and charge, tissue pore density, extracellular matrix structure, hydrostatic gradients, and the distribution of antigen and Fc receptors in tissue[50]. We showed previously that within 24 h after a single 10 mg/kg subcutaneous dose of the bNAb cocktail used here, bNAbs were detectable in all tissue homogenates tested at a

median concentration of 174.5 ng/ml (range 50–791 ng/ml), and in most cases had measurable neutralizing activity in vitro[12]; however, no time points before 24 h were tested. Future studies of PEP with monoclonal antibodies should perform longitudinal tissue PK studies during the first hours after treatment to facilitate comparison with small-molecule ART regimens and to optimize treatment strategies.

Although bNAb treatment at 48 h did not prevent persistent infection, viremia was attenuated in the majority of animals. Regardless of treatment regimen, infants that had tightly controlled or undetectable infection did not mount adaptive immune responses. The lack of viral resurgence after CD8+ T cell depletion in vivo supported the idea that tight control was maintained independent of CD8+ immunity, contrasting with previous reports in adult macaques[30,44]. In the current study, tight control may be actively maintained by other means, which may include localized innate immunity, low fitness of the transmitted/founder virus, cell-intrinsic resistance to infection, sequestration in distinct anatomical sites, or a combination of these mechanisms. Alternatively, tight control may simply be a consequence of abortive infection due to potent blockade of establishment of a persistent replication-competent reservoir, with the rare detectable vDNA likely being defective. We attempted to distinguish these two possibilities using a quantitative viral outgrowth assay. However, because a lack of inducible virus in tight controllers does not necessarily indicate the absence of a replication-competent reservoir in vivo, we cannot determine whether tight control is an active process on the basis of viral outgrowth data alone. Understanding the longevity of the earliest seeding events and kinetics of persistent reservoir establishment is critical to elucidating the mechanism of tight control after early intervention.

Finally, our findings support the concept that maximizing the effective concentration and persistence of passively delivered bNAbs is critical for reducing viral spread after exposure. In a study that used a cocktail of less-potent bNAbs for PEP in newborns, only partial control of viremia was observed[51], and we demonstrate here that the required antibody concentration is reduced and window of opportunity for intervention is extended by the greater potency of the bNAbs used. These studies also revealed that a single high dose of bNAbs resulted in longer antibody persistence in vivo than a 4-dose regimen—even for PGT121, which had no Fc modifications to extend half-life in either regimen. The strong inverse association between ADA responses and bNAb half-life suggests that drug immunogenicity has an important effect on PEP efficacy. Indeed, the single bNAb treatment regimen minimized ADA and was therefore sufficient to blanket the time of greatest vulnerability for viral spread. Further studies are needed to test whether increasing bNAb persistence and titer in vivo, either by single-dose treatment or Fc engineering, could improve outcomes when therapy is initiated 48 h or later after exposure. Because the studies described here focused on SHIV infection at one month of age, during a time of rapid immune maturation, it will be important to test these concepts in newborn macaques to understand whether the immunological milieu contributes to control. Combination therapy using both bNAbs and ART may also be a highly effective approach and should be explored, especially for newborns exposed to HIV infection during birth or breastfeeding.

## Methods

**Ethics statement.** Macaque studies were performed at the Oregon National Primate Research Center (ONPRC) in Beaverton, OR, USA in compliance with all ethical regulations for animal testing and research. The ONPRC is accredited by the American Association for the Accreditation of Laboratory Animal Care (AAALAC) International, and adheres to the Guide for the Care and Use of Laboratory Animals and the U.S. Public Health Service Policy on the Humane Care and Use of Laboratory Animals. The study protocol was approved by the Oregon Health & Science University (OHSU) West Campus Institutional Animal Care and Use Committee (IACUC).

**Animal model.** For viral exposure and treatment studies in one-month-old infant rhesus macaques (*Macaca mulatta*), 31 male and female macaques were obtained from the breeding colony at one week of age and raised in the ABSL-2 infant nursery for 3 weeks, during which time they were adapted to formula feeding. Infants were then transferred to ABSL-2+containment for study procedures involving SHIV challenge. Animals were randomly assigned to study groups as they were born, and were excluded from the study if the animal, or its sire or dam, could not be confirmed negative for *Mamu*-B*08 and -B*17 MHC Class I alleles, which are associated with spontaneous lentiviral control[52,53]. Group sizes of six had previously been shown sufficient for statistically distinguishable measurements of plasma and cell-associated viral loads at 6 months as the primary study outcome for antibody treatment. Animals were housed in age-matched pairs throughout the study, and were monitored for clinical signs of disease by regular evaluation of body weight, peripheral lymph node size, appetite, behavior, and stool quality. Animals were euthanized under IACUC guidelines using standard methods consistent with the recommendations of the American Veterinary Medical Association (AVMA) Guidelines for Euthanasia[54].

**SHIV virus challenge.** For Group 2A, a challenge stock of SHIV$_{SF162P3}$ was obtained from the NIH AIDS Reagent Program, Division of AIDS, National Institute of Allergy and Infectious Diseases, National Institutes of Health ("SHIV$_{SF162P3}$ (NIH)", catalog number 6526; contributors J. Harouse, C. Cheng-Mayer, and R. Pal). Virus was diluted 4-fold in DMEM media just prior to oral challenge. Each animal received a total of 0.5 ml (885 TCID$_{50}$ measured in rhesus PBMC) of cell-free virus by swallowing, given as two 1 ml doses of diluted virus 15 min apart. For Groups 1, 2B, 3, and 4, a challenge stock of SHIV$_{SF162P3}$ virus ("SHIV$_{SF162P3}$ (OHSU-2017)") was generated in activated, magnetically-enriched CD4+ macaque splenocytes inoculated with a stock of SHIV$_{SF162P3}$ obtained from the NIH AIDS Reagent Program (catalog number 6526; contributors J. Harouse, C. Cheng-Mayer, and R. Pal). See next section for detailed virus production methods. Cell-free viral culture supernatant was harvested on day 7, and aliquots were stored in the liquid nitrogen gas phase until use in challenge experiments. Virus stock titer was quantified in vitro (see "Quantitation of Viral Stock Titer" section below). Animals received a total of 2 ml (4.1 × 10$^4$ TCID$_{50}$ measured in rhesus PBMC) of undiluted cell-free virus by swallowing, given as two 1 ml doses either 15 min apart (Group 2B) or 4 h apart (Groups 1, 3, 4). Virus aliquots were transported on dry ice and thawed at room temperature or in hand just prior to animal challenge.

**SHIV stock production.** Spleens and blood from naïve rhesus macaques were processed to generate single cell suspensions, which were cryopreserved in the liquid nitrogen gas phase. To generate a stock of SHIV$_{SF162P3}$ for use in animals, a small-scale virus expansion was first performed using SHIV$_{SF162P3}$ stock obtained from the NIH AIDS Reagent Program (catalog number 6526) as inoculum; the day 7 supernatant of this small culture was subsequently used to inoculate a large-scale culture for virus stock production. For the small-scale expansion, a total of 8 × 10$^7$ PBMC from three different rhesus macaques were thawed and rested separately overnight, then enriched for CD4+ cells by MACS using NHP CD4 MicroBeads (Miltenyi). CD4-enriched cells were pooled and activated for 24 h in R15-100 media (RPMI1640, 15% FBS, 100 U/ml IL-2) containing a stimulation cocktail of Staphylococcal enterotoxin B (2 µg/ml, Toxin Technologies) and antibodies against CD3 (300 ng/ml, clone CD3-1, Mabtech), CD28 (1.5 µg/ml, clone L293, BD Biosciences), and CD49d (1.5 µg/ml, clone 9C10, BD Biosciences). The next day, cells were washed and incubated for 2 more days in fresh R15-100 media. A total of 2 × 10$^7$ cells were then spinoculated for 2 h at 1600 × $g$ at RT with 50 µL of SHIV$_{SF162P3}$ virus from an aliquot that had previously experienced one freeze-thaw cycle. Cells were washed the next day to remove free virus, and cultures were maintained by refreshing half of the media every other day until supernatant was harvested and banked on day 7. Viral titer was quantified as described below. For the large-scale SHIV stock production, 2.5 × 10$^9$ rhesus splenocytes from a single animal were thawed and rested overnight, then subjected to CD4+ enrichment by MACS and activation in vitro, as described above. Spinoculation was performed using the small-scale SHIV supernatant as inoculum, at an MOI = 0.08 as measured by TCID$_{50}$/ml in TZM-bl cells (NIH AIDS Reagent Program, catalog number 8129). Virus cultures were maintained until supernatant was harvested on day 7, aliquoted, and cryopreserved in liquid nitrogen for use in vivo. This stock is referred to as "SHIV$_{SF162P3}$ (OHSU-2017)". Viral titer was quantified as described below.

**Quantitation of viral stock titer.** TCID$_{50}$/ml was measured in rhesus PBMC using an assay adapted from Ranajit Pal (ABL, personal communication). Briefly, the viral stock was serially diluted 4-fold in a 96-well plate, beginning with a 1:2 dilution, making seven replicates per dilution step. Viral dilutions were incubated with 2 × 10$^5$ PHA-stimulated naïve rhesus PBMC per well for 7 days at 37 °C under 5% CO$_2$. Supernatants were then harvested and the presence or absence of virus in

each well was determined by SIV p27 ELISA (RETROtek SIV p27 Antigen ELISA, ZeptoMetrix Corporation, Buffalo, NY). The positive cutoff value was determined by dividing the average $OD_{450}$ of the lowest SIV p27 standard in the ELISA plate (63 pg/ml) by 2, and adding 0.05. $TCID_{50}$ was calculated using the Spearman-Karber method. For $SHIV_{SF162P3}$ (OHSU-2017), the challenge dose was chosen based on the results of in vivo titration by oral inoculation in infant rhesus macaques.

**Monoclonal antibodies**. IgG1 antibodies VRC07-523, VRC07-523LS, and PGT121[31,32] were produced by transient transfection in Expi293F cells (Thermo-Fisher Scientific, Inc.) and affinity purified on Protein A columns to >95% purity. Antibodies were formulated in a cocktail and delivered subcutaneously around the dorsal cervical and thoracic regions according to the dosing regimens described in the text. For multiple-dose treatments, each dose was delivered at one site per day; the same or a different site may have been used on subsequent days.

**Antiretroviral therapy (ART)**. Animals in Group 4 were given daily treatment with a triple-drug ART regimen[34]. The drugs were coformulated in a cocktail with 5.1 mg/ml tenofovir disoproxil fumarate (TDF, Gilead Sciences, Inc.), 40 mg/ml emtricitabine (FTC, Gilead Sciences, Inc.), and 2.5 mg/ml dolutegravir (DTG sodium salt and free base, Shanghai Medicilon, Inc.) dissolved in 15% Kleptose HPB Parenteral grade (Roquette) in sterile water, adjusted to pH 4.2, sterile filtered and stored frozen at −20 °C. The cocktail was administered subcutaneously at 1 ml/kg each day for 21 days, beginning 48 h after SHIV exposure.

*Blood and tissue harvest and processing*: Peripheral blood was collected into EDTA blood tubes prior to virus exposure on the day of challenge, on days 4, 7, 10, and 14, and weekly thereafter. Blood tubes were centrifuged at 1850 rpm for 25 min at 4 °C with the brakes off in order to separate plasma from cells. The plasma supernatant was pipetted off and aliquots were stored at −80 °C. The remaining blood fraction was resuspended in sterile PBS to double the original volume, and PBMC were isolated by centrifugation in SepMate tubes (StemCell Technologies) over Lymphocyte Separation Medium (Corning). For vDNA detection by quantitative PCR as described below, $3 \times 10^6$ PBMC were pelleted at ~20,000 × g in a benchtop microcentrifuge and frozen at −80 °C; any remaining PBMC were cryopreserved in liquid nitrogen. Inguinal or axillary lymph nodes were biopsied at various times post-exposure, and single cell suspensions were made by crushing the tissue through a 100 μm strainer. Lymph node cells were then frozen as pellets of $3 \times 10^6$ cells at −80 °C for viral quantitation, and any remaining cells were cryopreserved in liquid nitrogen. At necropsy, blood, cerebrospinal fluid (CSF), and a panel of up to 31 solid tissues were harvested (see Supplementary Tables 2–6). CSF was collected into a 2 ml vial and stored at −80 °C. From each solid tissue, 100 μg samples were excised and frozen at −80 °C in 2 ml tubes pre-filled with 1.4 mm zirconia beads (Spex SamplePrep) to facilitate tissue homogenization with a bead beater, nucleic acid extraction, and vDNA detection by quantitative PCR. For spleen and mixed mesenteric lymph nodes, any remaining tissue was processed to make single cell suspensions, which were cryopreserved in liquid nitrogen for use in viral outgrowth assays.

*Viral nucleic acid detection in plasma, cells, and tissue homogenates*: Nucleic acid from plasma, CSF, or peripheral blood mononuclear cells (PBMC) was purified using a Maxwell 16 instrument (Promega, Madison, WI) per the manufacturer's protocol, using the LEV Viral Nucleic Acid Kit for plasma and CSF and the LEV Whole Blood Nucleic Acid Kit for cells. SHIV viral RNA in plasma and CSF was measured by quantitative RT-PCR with a detection limit of 50 copies/ml using a method developed by Cline et al.[55] with minor modifications to the master mix to increase sample input. SHIV vDNA in cellular DNA from PBMC or lymph node biopsy pellets was measured using quantitative PCR using Fast Advanced Mastermix on an Applied Biosystems QuantStudio 6 Flex instrument (Life Technologies, Carlsbad, CA). Reactions were performed with 2 μg nucleic acid input for 45 cycles using the FAST cycling protocol (95 °C for 1 s, 60 °C for 20 s) in a 30 μl reaction volume. Virus copy numbers were estimated by comparison to a linearized pBSII-SIVgag standard curve and calculated per cell equivalent using the input nucleic acid mass and by assuming a DNA content of 6.5 μg per $10^6$ cells, with a detection limit of 2 copies/μg DNA or 10 copies/$10^6$ cells. Primers and probe used for plasma and PBMC assays were those described by Cline et al.[55]: SGAG21 forward (GTCTGCGTCATPTGGTGCATTC), SGAG22 reverse (CACTAGKTGT CTCTGCACTATPTGTTTTG), and pSGAG23 (5′-(FAM)-CTTCPTCAGTKTG TTTCACTTTCTCTTCTGCG-(BHQ1)-3′). vDNA was measured in necropsy tissue samples using an ultrasensitive nested quantitative PCR assay[44] targeting a highly conserved region of gag in SIV and SHIV with a detection limit of 0.02 copies/μg DNA or 1 copy/$10^7$ cells. Primers used for pre-amplification of vDNA were SIVnestF01 (GATTTGGATTAGCAGAAAGCCTGTTG) and SIVnestR01 (GTTGGTCTACTTGTTTTTGGCATAGTTTC), and primers for quantitative PCR were SGAG21 forward, SGAG22 reverse, and pSGAG23 as described above. Samples were heated at 95 °C for 5 min and then put on ice. Each sample was assayed in 12 replicates of 5 μl each. In order to assess PCR reaction efficiency, 10 copies of DNA containing the SIV gag target sequence were spiked into two of the reactions. None of the tested DNA samples showed significant amplification inhibition, defined as a 5-cycle delay relative to the amplification kinetics of reactions containing solely 10 copies of standard. The first round of amplification was performed in 12 cycles (95 °C for 30 s, 60 °C for 1 min) in a 50 μl reaction

volume. For each pre-amplified replicate sample, 5 μl was used as input into quantitative PCR using the Fast Advanced Mastermix in the QuantStudio 6 Flex instrument. Reactions were performed for 45 cycles using the FAST cycling protocol described above in a 30 μl reaction volume. Virus copy numbers were calculated from the frequency of positive replicates using the Poisson distribution and expressed as copies per μg DNA, or as copies per cell equivalent by assuming a DNA content of 6.5 μg DNA per $10^6$ cells. Staff members performing the viral RNA and DNA assays were blinded to the experimental groups and conditions for all samples tested.

**Flow cytometry and $CD4^+$ T cell counts**. For each sample, 100 μl of whole blood was transferred from an EDTA blood collection tube into a cluster tube and washed twice with 1 ml PBS, aspirating and vortexing between washes. Surface stain antibodies and live/dead dye (Live/Dead Fixable Yellow Dead Cell Stain Kit, Life Technologies L34968, 0.1 μl/test) were then added, and samples were vortexed and incubated for 30 min at room temperature in the dark. Red blood cells were lysed with 1 ml of 1X FACSLyse (BD Biosciences 349202) for 8 min, followed by three washes in FACS buffer (PBS, 10% FBS, 1 mM EDTA). Samples were fixed in 100 μl of 2% paraformaldehyde. The following antibodies were used for surface staining: CD45 PE-Cy7 (clone D058-1283, BD Biosciences 561294, 0.1 μl/test), CD3 Alex-aFluor 700 (clone SP34-2, BD Biosciences 561805, 1 μl/test), CD4 APC (clone M-T466, Miltenyi 130-113-250, 2.5 μl/test), CD8 Pacific Blue (clone DK25, Agilent Technologies PB98401-8, 2.5 μl/test), CD28 PE (clone 28.2, BD Biosciences 556622, 4 μl/test), CD95 FITC (clone DX2, BD Biosciences 556640, 5 μl/test), and CD20 PerCP-Cy5.5 (clone L27, BD Biosciences 340954, 1 μl/test). Prior to flow staining, a complete blood count (CBC) was performed on an aliquot of blood from the same day to obtain the absolute lymphocyte count per μl blood. The number of $CD4^+$ T cells per μl blood was calculated by multiplying the absolute lymphocyte count by the percentage of lymphocytes that were viable and $CD45^+$ $CD3^+$ $CD8^-$ $CD4^+$. See Supplementary Fig. 8 for an example of the gating strategy used.

*$CD8\alpha$ depletion and $CD8^+$ T cell counts*: To evaluate the contribution of $CD8^+$ T cell immunity to control of viremia during post-acute infection in a subset of SHIV-exposed and bNAb-treated animals, depleting anti-CD8α antibody (mouse/rhesus CDR-grafted IgG1, clone M-T807R1, NHP Reagent Resource, NIH) was administered subcutaneously in four doses. The initial dose was 10 mg/kg and three subsequent doses, 5 mg/kg each, were given 4, 8, and 11 days after the initial dose. Blood was drawn weekly thereafter, and peripheral $CD8^+$ T cell counts were monitored by complete blood counts in conjunction with flow cytometry using the staining protocol described above for $CD4^+$ T cells. The number of $CD8^+$ T cells per μl blood was calculated by multiplying the absolute lymphocyte count by the percentage of lymphocytes that were viable and $CD45^+$ $CD3^+$ $CD8^+$ $CD4^-$. See Supplementary Fig. 8 for an example of the gating strategy used.

*CD20 depletion*: To evaluate the contribution of B cell immunity to viral control during chronic infection, animals 36505 and 36557 in Group 2B received a single dose of depleting anti-CD20 antibody (mouse/rhesus CDR-grafted IgG1, clone 2B8, NHP Reagent Resource, NIH) administered subcutaneously at 50 mg/kg.

*Single-genome analysis (SGA)*: Viral RNA was isolated from plasma or viral stock samples using the QIAamp Viral RNA Mini kit (Qiagen, Valencia, CA), and cDNA was generated with BGenv3out (GGCCTCACTGATACCCCTACC) specific 3′ primer using the SuperScript III first-strand synthesis system (Invitrogen, Carlsbad, CA). cDNA was subjected to limiting dilution such that <30% of wells yielded a PCR product after single-genome amplification. Single-genome amplification (SGA) of full-length gp160 Envelope was performed according to the CHAVI standard operating procedure[56] using high-fidelity platinum Taq (Invitrogen) and the following primer sets. First-round primers for gp160 SGA were BGenv5out (GCTATACCGCCCTCTAGAAGC) and BGenv3out (GGCCTCACTGATACCCCTACC). The first-round thermocycler program was (94 °C × 2 min, [94 °C × 15 s, 58 °C × 30 s, 68 °C × 4 min] × 35 cycles, 68 °C × 15 min). Second-round primers for nested amplification were P3envB5in_NheI (GATCGCTAGCGTATGGGTCACAGCT) and P3envB3in_MLuI (GATCGACGCGTATCCATATTGTAGGT); these primers introduced the restriction sites for NheI and MLuI to enable insertion of the gp160 product into the pEMC* vector for downstream pseudovirus construction. The second-round thermocycler program was (94 °C × 2 min, [94 °C × 15 s, 58 °C × 30 s, 68 °C × 4 min] × 45 cycles, 68 °C × 15 min). Upon satisfying the SGA criteria for <30% positive results by gel electrophoresis, wells with a single genome were identified by Sanger sequencing analysis (elimination of sequences with early stop-codons or with double peaks) and the second-round PCR was repeated on the positive samples. The resulting 2.5 kb fragments were purified on a 1% agarose gel and cleaned up using the Wizard SV Gel and PCR Cleanup System kit. DNAs were digested with MLuI and NheI, then ligated into the pEMC* vector, which had been previously digested with MLuI and NheI and SAP treated using exoSAP-IT (Affymetrix, Santa Clara, CA). Ligation was performed using the Thermo RapidDNA Ligation Kit. The resulting gp160/pEMC* vector was transformed into competent E. coli using the MAX Efficiency Stbl2 Competent Cell Kit and grown overnight at 30 °C. Individual colonies were re-streaked and again grown overnight at 30 °C to obtain clonal populations, which were then grown up in culture. To confirm presence of gp160 in the vector, colony PCR was performed using the Promega GoTaq Flexi DNA Polymerase kit and primers SK1 (GATCCTT AAGGCAGCGGCAGAAGAA) and SK6 (GATCGTGTATGGCTGATTAT

GATGAT). The thermocycler program was (95 °C × 5 min, [95 °C × 45 s, 60 °C × 45 s, 72 °C × 2.5 min] x 35 cycles, 72 °C × 10 min). Products were run on a 1% agarose gel to confirm the presence of bands, and DNA was extracted from the confirmed cultures using the Promega PureYield Plasmid Miniprep System. To sequence the gp160, the following primers were used: 218 (ATCATTAC ACTTTAGAATCGC), ED5P3mod (ATGGGATCAAAGTCTAGAGCCCATGTG), KK1 (GCACAGTACAATGTACACATGGAA), env8R (CACAATCCTCGCT GCAATCAAG), env6For (GAATTGGATAAGTGGGCAAG), SK5 (GATC GCCGTGAATTTAAGGGACGCTG), and SK6 (GATCGTGTATGGCTGA TTATGATGAT). SHIV gp160/pEMC* vector DNA was stored at −20 °C until use.

*Env-pseudovirus construction*: SHIV gp160/pEMC* vector DNA was co-transfected with pSG3ΔEnv HIV-1 backbone DNA into 293T cells (European Collection of Authenticated Cell Cultures, Sigma catalog number 12022001) using the jetPEI (Polyplus) transfection reagent. Supernatant containing pseudovirions was harvested after 2–3 days and frozen in 1 ml aliquots at −80 °C. Pseudovirus stocks were titrated in TZM-bl cells (NIH AIDS Reagent Program, catalog number 8129) to determine the virus dilution required for 200,000 relative light units (RLU).

*Env-pseudovirus neutralization assay*: To determine whether emerging clones in bNAb-treated animals with breakthrough viremia were escape variants, a panel of bNAbs recognizing different epitopes were assayed in duplicate for neutralization against single round of entry $SHIV_{SF162P3}$ Env-pseudoviruses using TZM-bl reporter cells in which infection drives luciferase expression (NIH AIDS Reagent Program, catalog number 8129)[57]. Briefly, antibodies were serially diluted 3-fold in complete DMEM media (DMEM, 10% FBS, L-glutamine) and incubated with pseudovirus for 1 h at 37 °C in 96-well flat bottom plates. TZM-bl cells were harvested, mixed with DEAE dextran (7.5 μg/ml) to enhance viral uptake, and 10,000 cells were added to each well containing antibodies and virus. Additional control wells containing cells and virus only (no antibody) and cells only (no virus, no antibody) were included on each plate. Plates were incubated at 37 °C and 5% $CO_2$ for 48–72 h. Bright-Glo (Promega) luciferase substrate was added to each well, and luciferase activity (relative light units, RLU) was measured on a luminometer. Wells containing only cells and virus defined 100% RLU signal, and cells-only wells defined 0% RLU signal. To calculate neutralization potency, the RLU in each well was divided by the RLU in the virus-only wells to give the percentage of viral infection not neutralized by antibody. This % RLU was plotted against antibody concentration to generate a dose-response curve for each antibody, from which 50% neutralization titer ($IC_{50}$) could be interpolated.

*Pharmacokinetic determination*: Plasma bNAb levels were quantified using plates coated with either RSC3[58] for specific detection of VRC07-523 or ST0A9[59] for specific detection of PGT121. Nunc MaxiSorp (Thermo Fisher) plates were coated overnight with 200 ng/well of RSC3 in PBS, washed with PBST five times, and blocked with TBST with 5% milk and 2% BSA for 1 h at RT. Serial dilutions of all samples were plated in duplicate. Each bNAb (for standard curves) and positive and negative controls were included on each plate. Plasma was incubated for 1 h at RT, followed by a PBST wash. Bound bNAbs were probed with a horseradish peroxidase-labeled goat antihuman IgG (1:5000 dilution; Jackson Laboratories) for 30 min at RT. The plate was washed and TMB (Pierce) substrate was added. Once color was developed stopping buffer was added and the optical density at 450 nm was read. GraphPad Prism and Microsoft Office software was used to plot standard curves and calculate bNAb concentrations.

*Measurement of ADA responses*: ADA responses were evaluated as follows. Plasma from macaques that had been administered PGT121 and VRC07-523/ VRC07-523LS were diluted with PBS containing 5% skim milk, 2% BSA and 0.05% Tween 20. Five-fold serial dilutions ranging from 1:50 to 1:781,250 of these plasma were then added in duplicate wells to 96-well ELISA plates coated with 2 μg/ml of either PGT121 or VRC07-523/VRC07-523LS. The plate was incubated for 1 h at room temperature followed by a PBS-T (PBS with 0.05% Tween-20) wash. Bound monkey IgGs were then probed with a horseradish peroxidase (HRP)-conjugated anti-monkey IgG, Fc-specific (Southern Biotech) for 30 min at room temperature. The plate was then washed and SureBlue TMB (Kirkegaard & Perry Laboratories, Gaithersburg, MD) substrate was added. Once color was developed (typically 15–20 min), stopping buffer (1 N $H_2SO_4$) was added and the optical density at 450 nm was read. Endpoint titer was calculated by determining the lowest dilution that had optical density greater than five-fold of that in the background wells.

*Viral outgrowth assay*: To measure inducible replication-competent virus in tissues, the TZA assay was used as described[37] with the following modifications. Single cell suspensions were generated from spleen or mesenteric lymph nodes and cryopreserved in the liquid nitrogen gas phase. Aliquots of approximately $5 \times 10^7$ cells were thawed and rested overnight, then positively enriched for $CD4^+$ cells by magnetic-activated cell sorting (MACS) using NHP CD4 MicroBeads (Miltenyi). Bulk CD4-enriched cells were divided into two equal portions, and stimulated in vitro for 5 days in R15-100 media (RPMI1640, 15% FBS, 100 U/ml IL-2) using CD2/CD3/CD28 T cell stimulation beads (Miltenyi) at a 1:1 cell-to-bead ratio under two different culture conditions as follows. In the first condition, to block de novo cycles of replication during in vitro stimulation, tenofovir disoproxil fumarate (TDF, NIH AIDS Reagent Program, NIAID, NIH, catalog number 10198) was reconstituted in sterile PBS and added to cultures at a final concentration of 10 μM. In the second condition, TDF was omitted from the stimulation culture to permit de novo viral replication and thus increase assay sensitivity for detecting small quantities of inducible virus in the sample. Because this condition permits

additional infection during culture, the resulting viral outgrowth data overestimates the actual frequency of cells harboring inducible virus, and is therefore not strictly quantitative. On day 3 of stimulation, half of the cell culture supernatant volume was removed and replaced with fresh media. On the morning of the assay, $1 \times 10^4$ TZM-bl cells were plated into each well of a 96-well culture-treated flat bottom assay plate and allowed to adhere for 4–6 h. Stimulated CD4-enriched cell samples were then plated in quadruplicate on top of the TZM-bl cells in a fourfold dilution series starting with $2.5 \times 10^5$ cells/well. To detect the presence of infectious virus in each well, Bright-Glo (Promega) was added after 48 h and luciferase activity was read out on a luminometer. Wells were considered positive for viral outgrowth if the luciferase signal exceeded the average plus 3 standard deviations of 12 replicate wells containing only TZM-bl cells (NIH AIDS Reagent Program, catalog number 8129). The number of infectious units per million $CD4^+$ cells (IUPM) in each sample was calculated using the IUPMStats v1.0 Infection Frequency Calculator available at http://silicianolab.johnshopkins.edu/.

*Enzyme-linked immunosorbent assay (ELISA)*: Antibody responses were determined by measuring binding of plasma IgG to recombinant HIV-1 SF162 gp140 trimer. The gp140 trimer was produced as described[60] by transient transfection in Expi293F cells and purified over a *Galanthus nivalis* lectin-coupled agarose (GNA) column (Vector Laboratories, Burlingame, CA), followed by size exclusion chromatography on a Superdex 200 column (GE Healthcare Life Sciences) to separate trimer from monomer. The trimer fractions were pooled and concentrated, and aliquots were frozen at −20 °C. Half-well ELISA plates (Costar) were coated with SF162 gp140 trimer at 0.5 μg/ml in carbonate/bicarbonate buffer and incubated overnight at 4 °C. Plates were washed one time in wash buffer (0.1% Triton X-100 in 1X PBS) and blocked with 1% normal goat serum/5% nonfat dried milk in PBS for 1 h at RT. Serially diluted plasma samples were then incubated for 1 h at RT followed by three washes. Plates were probed with a horseradish peroxidase-conjugated goat anti-human IgG Fc fragment-specific polyclonal antibody (1:5000, Jackson ImmunoResearch) for 1 h at RT. After five washes, TMB substrate (SouthernBiotech) was added and incubated for 10 min before quenching with $H_2SO_4$. Optical density at 450 nm was read and 50% binding titers were calculated using GraphPad Prism and Microsoft Excel. Each sample was assayed in duplicate.

*Enzyme-linked Immunospot Assay (ELISPOT)*: T cell responses in blood and tissues were tested in an interferon-γ (IFN-γ) enzyme-linked immunosorbent spot assay. Briefly, $1 \times 10^5$ mononuclear cells were incubated with antigen in duplicate wells of an anti-NHP IFN-γ coated ELISPOT plate (Mabtech) overnight at 37 °C. To measure responses to Env, antigens tested were either HIV-1 SF162 Env (gp140) protein or pools of overlapping 15-mer peptides covering the length of the HIV-1 SF162 Env amino acid sequence. To detect anti-Gag responses, a pool of overlapping 15-mer peptides covering the entire SIVmac239 Gag open reading frame (ORF) was used. ELISPOT plates were then probed using a biotinylated anti-IFN-γ antibody followed by Streptavidin-Alkaline Phosphatase and developed with BCIP/NBT-plus substrate. Plates were read on an ELISPOT plate reader (Autoimmun Diagnostika GMBH). Cells incubated with Concanavalin A (Sigma-Aldrich) and Staphylococcal Enterotoxin B (Toxin Technologies) were used as positive controls, while antigens were omitted for a background control. The numbers of spots in duplicate wells were averaged and normalized to calculate the number of spot-forming cells per $1 \times 10^6$ cells. Wells with <50 spot-forming cells per $1 \times 10^6$ cells were considered negative. Positive responses were determined by using a one-tailed Student t test at α = 0.05, where the null hypothesis was that the number of spots in the treatment wells would be lower than or equal to the background. Values determined to be positive were reported as the average of duplicate test wells minus the average of all negative control wells.

*Evaluation of Pathology*: At necropsy, rhesus macaque tissues and body fluids were collected fresh, frozen, or fixed in 10% formalin and subsequently processed for histologic evaluation. Necropsies and microscopic evaluation of tissues were performed by veterinary pathologists. Pathogens were identified and confirmed by H&E morphology and then histochemical stains, immunohistochemistry (IHC), and PCR. Adenovirus, cytomegalovirus, and *Enterocytozoon bieneusi* were all visualized by IHC using the Vectastain ABC Kit, Peroxidase Standard (Vector Laboratories, Burlingame, CA, USA). Adenovirus was detected with a mouse anti-adenovirus monoclonal antibody (MAB8052, 1:500 dilution, EMO Millipore, Temecula, CA, USA). Cytomegalovirus was detected using an antibody kindly. pngted by Peter Barry (1:750 dilution) as previously reported[61]. *E. bieneusi* were visualized using a mouse anti-measles matrix protein monoclonal antibody (MAB8910, 1:1000 dilution, EMO Millipore, Temecula, CA, USA). *Spironucleus* species (sp) were confirmed in tissue by nested PCR amplification of small subunit ribosomal DNA using methods developed by Bailey et al.[62] (see Supplementary Table 13 for primer sequences). *Cryptosporidium* sp, SHIV giant cell disease, flagellated protozoa, *Pneumocystis* sp, *Malassezia* sp, and attaching and effacing *E. coli* were all diagnosed by H&E stain. Additional diagnostic methods included Gomori methenamine-silver stain for *Pneumocystis* sp and Periodic Schiff reaction for *Malassezia* sp. Intracellular agyrophilic bacteria were visualized by Warthin-Starry stain.

*Statistical analysis*: In Supplementary Fig. 3, the percentage of aviremic animals in each treatment group was compared with the untreated control group using a Log-rank test in which the threshold for significance was adjusted for multiple comparisons by the Dunnett-Hsu method. In the text discussing vDNA copies in tissues (related to Fig. 4), the ratio of DNA copies present in lymphoid

vs. GI tract tissues in viremic animals was calculated by dividing the average of the copy numbers present in lymphoid tissues for each viremic animal by the average of the copy numbers present in GI tract tissues for that animal ($n = 8$). The mean and 95% confidence interval of these ratios is reported in the text. A D'Agostino and Pearson normality test was used to determine whether the ratios were derived from a normally distributed data set. Because the data were normally distributed ($K^2 = 0.3989$, $p = 0.8192$), a one-sample $t$ test was used to test the hypothesis that the ratio of DNA copies present in lymphoid vs. GI tract tissues was equal to 1. Lymphoid tissues included all lymph nodes sampled (see Supplementary Table 1), as well as tonsil and spleen; GI tract tissues were duodenum, jejunum, ileum, cecum, colon descending, and rectum. For analysis of correlations between PVL AUC and vDNA in tissues (Supplementary Table 7), PVL data was censored after 10 weeks in order to permit comparison between animals that remained on study for different lengths of time. All animals in Groups 1, 2B, 3, and 4 were included in the analysis except for two animals, 36206 and 36207, which were excluded because they were sacrificed prior to 10 weeks. All 23 animals in Groups 1, 2B, 3, and 4 were included in the analysis of correlations between tissue vDNA and PVL peak or PVL final. Due to the skewed distribution of the data, nonparametric correlations were used. Statistical analyses were performed in GraphPad Prism 7 or in SAS 9.4.

**Reporting summary**. Further information on research design is available in the Nature Research Reporting Summary linked to this article.

## Data availability

All datasets analysed during this study are included in this published manuscript and its Supplementary Information. Data underlying Fig. 4 are provided in Supplementary Tables 2–6. Data underlying Figs. 2, 3, 5, and Supplementary Figs. 1–7 are provided as Source Data files. All other data are available from the corresponding author on reasonable request.

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

## Acknowledgements

We thank Dr. Ranajit Pal for discussions about quantitation of viral titer, and Drs. Gabriela Webb and Shaheed Abdulhaqq for discussions regarding virus stock production. We acknowledge technical assistance from Philip Barnette and William Sutton. We thank Dr. Rebecca M. Ducore, Lois M.A. Colgin and Heidi L. Pecoraro for performing necropsies and evaluating tissues for the study, and Dr. Peter Barry for kindly providing the cytomegalovirus detection antibody for pathology analyses. We acknowledge the veterinary and technical staff of the Division of Comparative Medicine at the Oregon National Primate Research Center for animal husbandry. This work was supported by NIH R01 HD080459 (N.L.H.), U42 OD023038 (M.K.A.), U42 OD010426 (M.K.A.), and P51 OD011092 (ONPRC), and by the intramural research program of the Vaccine Research Center, NIAID, NIH. M.B.S. was supported by NIH T32 AI007472.

## Author contributions

M.B.S., J.B.S., J.R.M., A.J.H., and N.L.H. designed the experiments. M.B.S. optimized SHIV stock production, performed viral outgrowth assays, and analyzed and interpreted data. T.C. managed tissue collection and databases, analyzed virology data, and performed antibody and viral outgrowth assays. D.C.M. designed and performed single genome cloning and bNAb escape studies. S.P. coordinated animal assignments, treatments, and procedures. M.B.S. and S.P. generated and titered the SHIV$_{SF162P3}$ virus stock. J.R. performed ELISPOT assays and quantified peripheral T cells. E.S.Y., A.P., and K.W. performed and interpreted quantitative ELISAs for bNAb pharmacokinetics. X.C. provided the antibodies for in vivo use and proteins for performing the quantitative ELISAs. D.S. and D.B. measured virus in plasma and tissues. H.H. and R.L. performed viral outgrowth and antibody binding and neutralization assays. M.F. collected samples from animals and performed T cell subset analyses. J.J.S. provided veterinary care and advice for animals. M.K.A. coordinated infectious disease resource animals. C.K. contributed molecular virology expertise and data analyses. B.P. performed statistical analyses. A.D.L. performed necropsies and described pathology. All authors discussed the results. M.B.S., A.J.H., and N.L.H. wrote the manuscript. N.L.H. supervised the research.

## Competing interests

The authors declare no competing interests.
