## [Peer Review File · Nature Communications]

REVIEWERS' COMMENTS:

Reviewer #2 (Remarks to the Author):

The authors revised the discussion enough to allow publication in my opinion.

Reviewer #3 (Remarks to the Author):

The reviewers did a thorough job addressing the reviewers comments and strengthening the manuscript based on the reviews.

One comment that I believe needs addressing in the manuscript is the point that in this model of peripartum HIV transmission, the infant monkeys were 1 mo of age before the virus dose was given. Since the immune system is rapidly changing in the neonatal period and infant monkeys mature more rapidly than humans, it could be significant that the infants were 1 mo of age as opposed to closer to birth. therefore this caveat should be pointed out in the manuscript.

NCOMMS-19-29159-T

REVIEWERS' COMMENTS:

Reviewer #2 (Remarks to the Author):

The authors revised the discussion enough to allow publication in my opinion.

Response: We appreciate the reviewer's approval.

Reviewer #3 (Remarks to the Author):

The reviewers did a thorough job addressing the reviewers comments and strengthening the manuscript based on the reviews.

One comment that I believe needs addressing in the manuscript is the point that in this model of peripartum HIV transmission, the infant monkeys were 1 mo of age before the virus dose was given. Since the immune system is rapidly changing in the neonatal period and infant monkeys mature more rapidly than humans, it could be significant that the infants were 1 mo of age as opposed to closer to birth. therefore this caveat should be pointed out in the manuscript.

Response: We thank the reviewer for bringing up this point, and agree that the animals' age is an important caveat for this model given the rapid immune maturation taking place in the newborn period. We have revised the manuscript discussion accordingly.